# Detection of Pneumothorax in Severe Acute Respiratory Distress Syndrome—Lung Ultrasound Pitfalls

**DOI:** 10.3390/diagnostics14020206

**Published:** 2024-01-18

**Authors:** Konrad Mendrala, Sylweriusz Kosiński, Tomasz Czober, Paweł Podsiadło, Szymon Skoczyński, Tomasz Darocha

**Affiliations:** 1Department of Anesthesiology and Intensive Care Medicine, Medical University of Silesia, 40-055 Katowice, Poland; 2Department of Intensive Interdisciplinary Therapy, Jagiellonian University Collegium Medicum, 31-008 Krakow, Poland; 3Department of Emergency Medicine, Jan Kochanowski University, 25-369 Kielce, Poland; 4Department of Lung Diseases and Tuberculosis, Faculty of Medical Sciences in Zabrze, Medical University of Silesia in Katowice, 40-055 Katowice, Poland

**Keywords:** ARDS, pneumothorax, lung ultrasound

## Abstract

Lung ultrasound is gaining popularity as a quick, easy, and accurate method for the detection of pneumothorax. The typical sonographic features of pneumothorax are the absence of lung sliding, the presence of a lung point, the absence of a lung pulse, and the absence of B-lines. However, we found that in some cases, each of these elements might be misleading.

The extensive range of diagnostic capability of ultrasound system has made the bedside ultrasound an indispensable diagnostic element [1]. Characterized by high sensitivity and specificity, the accuracy of ultrasound in detecting pneumothorax exceeds chest radiograph [2]. The detection of pleural sliding is fundamental, and a careful assessment should be based primarily on B-mode images. 

The mechanical ventilation of ARDS patients is likely to make it challenging due to high PEEP and small tidal volumes. Therefore, the absence of pleural sliding cannot be the only symptom determining the diagnosis of pneumothorax. In cases of diminished lung sliding, a high-frequency linear transducer should be used to better visualize the pleura. Alternatively, M-mode combined with real-time ECG may be useful to search for the so-called lung pulse. A “pseudo” lung pulse may result from intercostal muscle contraction in a spontaneously breathing patient, but unlike a real lung pulse, it starts above the pleural line, passes down crossing it, and does not correlate with the heart rate (Figure 1A; Appendix A) [3]. 

In hydropneumothorax, a typical lung point (Figure 1B; Appendix A) should be distinguished from a sign called the hydro-point (Figure 2; Appendix A), where the air/fluid border appears as the interposition between an anechoic space and a non-sliding A-pattern [4]. The observed phenomenon resembles a curtain sign, but unlike the typical one, it can be seen in various areas of the chest wall, not only the base. Also, searching for the lung point can be difficult, time-consuming, or even impossible when the whole lung is collapsed. 

According to the literature, “the presence of subpleural artifacts rules out pneumothorax in 100%”. We would rather agree on 99%. Comet tails artifacts may result from the presence of pleural adhesions in the pneumothorax chamber or ruptured bullous emphysema, which imitate the physiological connective tissue septa (Figure 3A,B; Appendix A) [5]. Also, subcutaneous emphysema is particularly disruptive to a lung ultrasound as it may obscure both normal structures and mimic other pathologies. In the area of subcutaneous emphysema, LUS may completely lose diagnostic value and should be interpreted with extreme caution. The presence of air in the subcutaneous tissue can mimic the A-profile as well as B-profile and subpleural consolidations (Figure 3C–E). For this reason, it is crucial to position the probe transversely to the ribs and correctly identify the anatomical landmarks.

## Figures and Tables

**Figure 1 diagnostics-14-00206-f001:**
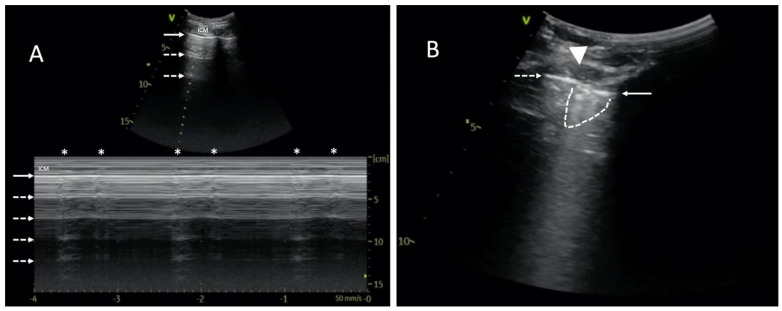
Point-of-care lung ultrasound of a 59-year-old male with severe ARDS, mechanically ventilated (Bilevel; FiO_2_ 60%; P_L_10 cmH_2_O; P_H_15 cmH_2_O). A left-sided pneumothorax occurred after pacemaker implantation. The study was performed using a 1.8–6 MHz convex probe. (**A**) Muscle contractions imitate lung pulse and hinder the diagnosis of pneumothorax. LUS examination revealed horizontal reverberation artifacts (dashed arrows). M-mode image shows a barcode sign with visible vertical artifacts (asterisks). Unlike a real lung pulse, the artifacts originate from the intercostal muscles (ICMs). Therefore, they start superficially to the pleural line and move down, passing the pleural line (solid arrow). Please see Appendix A. (**B**) Lung point is a pathognomonic sign of pneumothorax and can be fully seen in Appendix A. This static image shows both pleural laminae separated from each other. As a result of pneumothorax, only the parietal pleura is visible on the left side (indicated with a dashed arrow). On the right side, a part of the lung with fragmented pleural line (solid arrow) and subpleural consolidation (dashed line) can be seen. The contact area is called the lung point (triangle).

**Figure 2 diagnostics-14-00206-f002:**
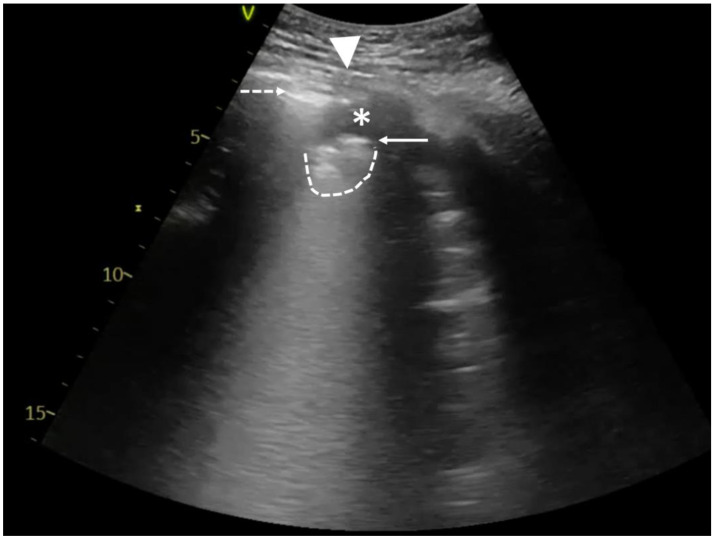
Point-of-care LUS of a 59-year-old smoker with COPD and severe ARDS. LUS was obtained with 1.8–6 MHz convex probe during vv-ECMO therapy and baby lung ventilation (Bilevel; FiO_2_ 40%; P_L_10 cmH_2_O; P_H_ 5 cmH_2_O; V_t_~100–150 mL). Considering the LUS images obtained in the previous days (bilateral consolidations), the A-profile on the left side of chest wall was unusual. A careful assessment with LUS revealed the hydro-point, where a pleural effusion in the left pleural cavity coexisted with pneumothorax. Pneumothorax is present on the left side of the image with a blurred line of the parietal pleura (dashed arrow), and no horizontal reverberations developed at the air/tissue boundary. On the right side of the image, fluid in the pleural cavity (asterisk) and irregular, fragmented pleural line (solid arrow) with the subpleural consolidation (dashed line) can be observed. The contact point of both is called hydro-point (triangle), one of the non-typical LUS images confirming the presence of pneumothorax. This dynamic sign, occurring suddenly and transiently on the ultrasound image, can be seen in Appendix A.

**Figure 3 diagnostics-14-00206-f003:**
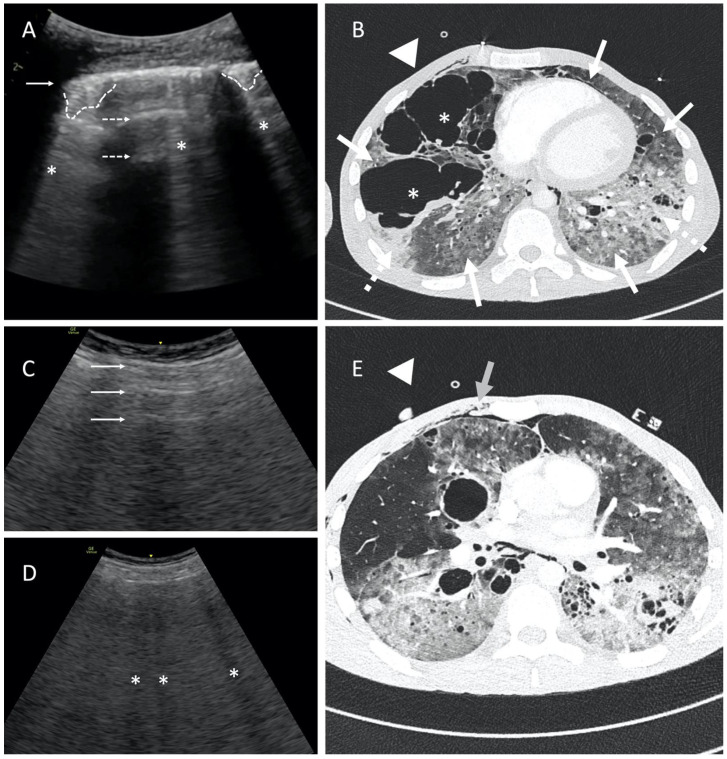
Point-of-care LUS image of 24-year-old male with human immunodeficiency virus (HIV) infection, ARDS, and suspected atypical bacterial pneumonia. Patient was ventilated with guaranteed tidal volume (Bilevel Volume Guarantee; FiO_2_ 100%; V_t_ 400 mL; PEEP 6 cmH_2_O; PIP 30 cmH_2_O). The study was performed using a 1.8–6 MHz convex probe. Presence of comet tail artifacts imitating B-lines carries a high risk of improper pneumothorax exclusion. (**A**) A-profile with blurred, fragmented pleural line (solid arrow), horizontal reverberations (dashed arrows), and disseminated subpleural consolidations (dashed lines). Note comet tail artifacts emerging most likely from subpleural structures (asterisks). Please refer also to Appendix A. (**B**) CT image shows advanced bullous emphysema (asterisks), massive consolidations (dashed arrows), and diffused ground-glass opacification (solid arrows). The probe position is marked with a triangle. (**C**) LUS image of horizontal artifacts similar to A-line (solid arrows) and (**D**) hypoechoic vertical artifacts mimicking B-lines (asterisks) in some way, limiting the utility of ultrasound in pneumothorax detection. Note the ribs and rib shadows are not visible; the pleural line is not visible. (**E**) The LUS image corresponds to subcutaneous emphysema in the upper part of the chest wall in CT scan (solid arrow). The probe position is marked with a triangle.

## Data Availability

The data presented in this study are available on request from the corresponding author.

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
