# Peer review of "Detection of Pneumothorax in Severe Acute Respiratory Distress Syndrome—Lung Ultrasound Pitfalls"

_diagnostics, 2024, doi:10.3390/diagnostics14020206_

Round 1

Reviewer 1 Report

Comments and Suggestions for Authors

The authors review some unique and interesting lung ultrasound cases, highlighting some of the potential pitfalls that could be misinterpreted.    As use of lung ultrasound continues to increase, it is could to review and explain the findings in very complex cases.    Below are a few points for revision.

1)     I would consider revising the title leaving out the ARDS component, as that does not seem to be integral to the cases.   Because the case with Bullous emphysema is included, I would say it is more like “Assessing for pneumothorax in complex pulmonary cases- diagnostic pitfalls.

2)     The videos are so essential,  I would make reference to the videos in the explanations of the stills shots (figure 1, etc).   This is because in the explanations of the still shots (if seen alone) would be somewhat misleading.   For example, figure 1 part B, in a still shot of a pleural line, I don’t think you can ever say “the pneumothorax is visible.”    It is the lack of lung sliding on the video in that area that makes it “visible”

3)     With regard to the first case presented, I agree motion artifacts in M-mode are a pitfall of M-mode.  This can be from chest wall movement or due to the sonographer moving the transducer.   I have not seen it mistaken for lung pulse, but I guess it could it if is rhythmic enough.  I agree correlating it with HR vs movement is key.   I like the point about the   M-mode the motion artifacts, but rather than saying “passing through the pleural line” I would say the artifacts “start superficial to the pleural line”.    The reader could interpret passing through as passing through vertically or horizontally, it took me a minute to figure out what the authors meant. 

4)     The authors should include information about type of probe (type, frequency, model) that was used in each exam.

5)     The first case appears to have been done with a low frequency convex (?curvilinear based on the appearance of the footprint).   I think it is a problem that there is no mention of the role of a high frequency transducer such a linear array transducer when there is “questionable pleural sliding” as the authors mention, when done with a low frequency transducer.   A 2-D assessment with a high frequency is better than M-mode with a low frequency transducer.

6)     Case 2 has an interesting image; the pneumothorax seems to create an intra-thoracic curtain sign.   The caption describes a “reference image” but no other image is included.  If the authors want to use that term they should include an image.

7)     The figure 3 caption talks about part A being “high risk of improper pneumothorax exclusion” but they go on to say it is bullous emphysema….so it really is not improper pneumothorax exclusion.  The ultrasound findings match areas of abnormal lung with a bulla in the middle. 

8)     For 3 part C, I would highlight the lack of rib shadows as anatomic land marks.  For part E I would strength the statement to say, once subcutaneous emphysema is identified, lung ultrasound should not be used as it is very unreliable. 

Author Response

Dear reviewer, thank you kindly for your time, we sought to improve the manuscript according to your comments - most of which we fully agree with. 

1) I would consider revising the title leaving out the ARDS component, as that does not seem to be integral to the cases.   Because the case with Bullous emphysema is included, I would say it is more like “Assessing for pneumothorax in complex pulmonary cases- diagnostic pitfalls.

Thank you for this comment, all three cases involved patients with severe ARDS - the bullous emphysema in the third patient (24 year old!) was most likely a result of destruction of lung tissue by viral infection and bacterial coinfection. We propose the short and informative title "Detection of pneumothorax in severe ARDS - lung ultrasound pitfalls".

2)     The videos are so essential,  I would make reference to the videos in the explanations of the stills shots (figure 1, etc).   This is because in the explanations of the still shots (if seen alone) would be somewhat misleading.   For example, figure 1 part B, in a still shot of a pleural line, I don’t think you can ever say “the pneumothorax is visible.”    It is the lack of lung sliding on the video in that area that makes it “visible”

We agree with the reviewer's opinion, corrections have been made to the manuscript.

3)     With regard to the first case presented, I agree motion artifacts in M-mode are a pitfall of M-mode.  This can be from chest wall movement or due to the sonographer moving the transducer.   I have not seen it mistaken for lung pulse, but I guess it could it if is rhythmic enough.  I agree correlating it with HR vs movement is key.   I like the point about the   M-mode the motion artifacts, but rather than saying “passing through the pleural line” I would say the artifacts “start superficial to the pleural line”.    The reader could interpret passing through as passing through vertically or horizontally, it took me a minute to figure out what the authors meant.  

Done.

4)     The authors should include information about type of probe (type, frequency, model) that was used in each exam.

      Done. We suggest omitting the model name and manufacturer of the transducer.

5)     The first case appears to have been done with a low frequency convex (?curvilinear based on the appearance of the footprint).   I think it is a problem that there is no mention of the role of a high frequency transducer such a linear array transducer when there is “questionable pleural sliding” as the authors mention, when done with a low frequency transducer.   A 2-D assessment with a high frequency is better than M-mode with a low frequency transducer.

Amendments were made to the abstract.

6)     Case 2 has an interesting image; the pneumothorax seems to create an intra-thoracic curtain sign.   The caption describes a “reference image” but no other image is included.  If the authors want to use that term they should include an image.

      The term 'reference image' referred to a comparison with the other lung - we agree with the reviewer that this may be confusing and removed this fragment. Also, we added a comparison to the "curtain sign" in the abstract.

7)     The figure 3 caption talks about part A being “high risk of improper pneumothorax exclusion” but they go on to say it is bullous emphysema….so it really is not improper pneumothorax exclusion.  The ultrasound findings match areas of abnormal lung with a bulla in the middle.  

We agree with the reviewer; however, the risk of rupture is high and it potentially carries the risk of observing subpleural lesions within the pneumothorax. According to the radiological description and considering the subcutaneous emphysema, the presence of pneumothorax cannot be ruled out (visible loose "detached" fragment of the septum between the two bullae?) Due to the patient's death, a second CT was not performed. We do not have an alternative case therefore we suggest to leave it in the manuscript for its educational value.

8)     For 3 part C, I would highlight the lack of rib shadows as anatomic land marks.  For part E I would strength the statement to say, once subcutaneous emphysema is identified, lung ultrasound should not be used as it is very unreliable. 

We have added this sentence to the abstract and corrected the fig 3 caption.

Reviewer 2 Report

Comments and Suggestions for Authors

Dear authors, congratulations on this project to revise, with the assistance of images and videos, the pitfalls that may compromise the diagnosis of pneumothorax in severe ARDS patients.

You can find my suggestions below.

-In my humble opinion, the article should be introduced by a concise and direct abstract that outlines the state of the art and the purpose of your work. Subsequently, elaborate on the features and challenges of lung ultrasound in ARDS patients with pneumothorax, explaining and commenting on your images.

-The bibliography could be expanded.

-Please mention the so-called 'curtain sign' in relation to hydropneumothorax.

-In affiliations, the second entry is incomplete as it lacks the country of origin.

-On line 61, would you mean parietal pleura?

Author Response

Dear reviewer, thank you very much for your time and valuable comments, we sought to improve the manuscript according to the comments - most of which we fully agree with. We consider the form of this manuscript as "short communication", so it is difficult to cover all aspects of pneumothorax diagnosis. 

1) In my humble opinion, the article should be introduced by a concise and direct abstract that outlines the state of the art and the purpose of your work. Subsequently, elaborate on the features and challenges of lung ultrasound in ARDS patients with pneumothorax, explaining and commenting on your images.

We would be more than happy to thoroughly discuss the ultrasound diagnosis of pneumothorax along with potential challenges. However, as suggested by the Editor, the article was categorized in the "interesting images" section, which imposes a strict form (no manuscript body, abstract only with a maximum of 200 words, which we exceeded…). Therefore, we propose to keep it in this “simplified form” addressed to a more advanced audience - abstract to summarize and structure potential problems, video to show and discuss some details.

2)The bibliography could be expanded.

We are also limited by the short form of the manuscript. We have selected the most representative publications.

3)Please mention the so-called 'curtain sign' in relation to hydropneumothorax.

Done

4)In affiliations, the second entry is incomplete as it lacks the country of origin.

This has been corrected.

5)On line 61, would you mean parietal pleura?

Thank you kindly for pointing out this error, it has been corrected.

Reviewer 3 Report

Comments and Suggestions for Authors

Dear authors,

This article presents a critical perspective on the use of lung ultrasound for detecting pneumothorax, highlighting potential pitfalls and challenges in its interpretation. It introduces various aspects that question the reliability of certain sonographic features commonly associated with pneumothorax.

Strengths:

1. Highlighting Limitations: The article aptly points out that the absence of classic sonographic signs, such as pleural sliding or the presence of specific artifacts, may not always definitively indicate the presence or absence of pneumothorax. This critical viewpoint challenges the assumption of a straightforward diagnosis based on these features.

2. Contextual Challenges: Acknowledging the complexities arising from specific clinical scenarios, like mechanical ventilation in ARDS patients, the article emphasizes the limitations in relying solely on traditional ultrasound features for pneumothorax diagnosis. This contextual awareness adds depth to the discussion.

3. Differentiating Findings: It distinguishes between the typical lung point and the hydro-point in hydropneumothorax, providing valuable insights and images into differentiating features that might otherwise be conflated. This comprehensive examination provides a more detailed and precise understanding of the significance of the subject matter.

Weaknesses:

1. Limited Evidence Base: While the article brings attention to potential challenges and misleading aspects of sonographic findings, it might lack a robust empirical foundation. The claims made regarding the limitations could benefit from more extensive clinical studies or evidence to support them.

2. Potential Oversimplification: While it rightly questions the reliability of certain sonographic signs, the article might risk oversimplifying the role of lung ultrasound in diagnosing pneumothorax. This could lead to potential skepticism without offering a nuanced alternative or broader diagnostic considerations.

3. Limitations: The article primarily focuses on challenges and limitations, leaving out potential advancements or solutions that might address these concerns. A more balanced approach could involve discussing ongoing research or proposed improvements in ultrasound techniques for better accuracy. Another potential limitation could be the requirement to construct a succinct ending for this particular style of presentation.

Comments on the Quality of English Language

 Minor editing of English language required

Author Response

Dear reviewer, thank you very much for your time and valuable comments, we sought to improve the manuscript according to the comments - most of which we fully agree with. We consider the form of this manuscript as "short communication", so it is difficult to cover all aspects of pneumothorax diagnosis. 

Strengths:

  1. Highlighting Limitations:The article aptly points out that the absence of classic sonographic signs, such as pleural sliding or the presence of specific artifacts, may not always definitively indicate the presence or absence of pneumothorax. This critical viewpoint challenges the assumption of a straightforward diagnosis based on these features.
  2. Contextual Challenges:Acknowledging the complexities arising from specific clinical scenarios, like mechanical ventilation in ARDS patients, the article emphasizes the limitations in relying solely on traditional ultrasound features for pneumothorax diagnosis. This contextual awareness adds depth to the discussion.
  3. Differentiating Findings:It distinguishes between the typical lung point and the hydro-point in hydropneumothorax, providing valuable insights and images into differentiating features that might otherwise be conflated. This comprehensive examination provides a more detailed and precise understanding of the significance of the subject matter.

Thank you for highlighting the strengths of the manuscript.

Weaknesses:

  1. Limited Evidence Base:While the article brings attention to potential challenges and misleading aspects of sonographic findings, it might lack a robust empirical foundation. The claims made regarding the limitations could benefit from more extensive clinical studies or evidence to support them.

The manuscript deals with a very narrow topic and has been written for an experienced audience. The 'interesting image' (200 words) form limits the possibility of describing a complete step-by-step diagnosis of pneumothorax.

  1. Potential Oversimplification:While it rightly questions the reliability of certain sonographic signs, the article might risk oversimplifying the role of lung ultrasound in diagnosing pneumothorax. This could lead to potential skepticism without offering a nuanced alternative or broader diagnostic considerations.

The presented cases represent rare, but not incidental, clinical conditions. At the beginning of the abstract, it was pointed out that LUS is a very good tool for bedside detection of pneumothorax. We propose to leave the form of the manuscript unchanged.

  1. Limitations:The article primarily focuses on challenges and limitations, leaving out potential advancements or solutions that might address these concerns. A more balanced approach could involve discussing ongoing research or proposed improvements in ultrasound techniques for better accuracy. Another potential limitation could be the requirement to construct a succinct ending for this particular style of presentation.

Thank you for this valuable comment, as already mentioned we are significantly constrained by the MDPI guidelines for this kind of manuscript. We propose to leave it in its current form.

Round 2

Reviewer 2 Report

Comments and Suggestions for Authors

I have no further suggestions. Thank you.

Reviewer 3 Report

Comments and Suggestions for Authors

Dear authors,

The revisions made in this manuscript version are beneficial for enhancing comprehension among experienced sonographers. The picture possesses excellent quality. The work focuses on a very specific subject matter and is intended for a knowledgeable readership.

The additional information holds great potential for comprehending the underlying mechanisms of ultrasonic artifacts.

Comments on the Quality of English Language

Minor editing of English language is required.